# Machine learning–based new classification for immune infiltration of gliomas

**Feng Yuan**[1]\*, **Yingshuai Wang**[2], **Lei Yuan**[3], **Lei Ye**[1], **Yangchun Hu**[1]\*, **Hongwei Cheng**[1]\*, **Yan Li**[1]\*

**1** Department of Neurosurgery, The First Affiliated Hospital of Anhui Medical University, Hefei, Anhui, China, **2** Department of Internal Medicine III, University Hospital Munich, Ludwig-Maximilians- University Munich, Munich, Germany, **3** Department of Anesthesiology, The First Affiliated Hospital of Anhui Medical University, Hefei, Anhui, China

\* Tisizi0707@sina.com (YL); Hongwei.cheng@ahmu.edu.cn (HC); hmchyczsl@126.com (YH); fengyuan1993@smail.nju.edu.cn (FY)

## Abstract

### Background

Glioma is a highly heterogeneous and poorly immunogenic malignant tumor, with limited efficacy of immunotherapy. The characteristics of the immunosuppressive tumor microenvironment (TME) are one of the important factors hindering the effectiveness of immunotherapy. Therefore, this study aims to reveal the immune microenvironment (IME) characteristics of glioma and predict different immune subtypes using machine learning methods, providing guidance for immune therapy in glioma.

### Methods

We first performed unsupervised cluster analysis on the genes and arrays of 693 gliomas in CGGA database and 702 gliomas in TCGA database. Then establish and verify the classification model through Machine Learning (ML). Then, use DAVID to perform functional enrichment analysis for different immune subtypes. Next step, analyze the immune cell distribution, stemness maintenance, mesenchymal phenotype, neuronal phenotype, tumorigenic cytokines, molecular and clinical characteristics of different immune subtypes of gliomas.

### Results

Firstly, we divide the IME of gliomas in the CGGA database into four different subtypes, namely IM1, IM2, IM3, and IM4; similarly, the IME of gliomas in the TCGA database can also be divided into four different subtypes (IMA, IMB, IMC, and IMD). Next, based on ML, we developed a highly reliable model for predicting different immune subtypes of glioma. Then, we found that Monocytic lineage, Myeloid dendritic cells, NK cells and CD8 T cells had the highest enrichment in the IM1/IMD subtypes. Cytotoxic lymphocytes were highest expressed in the IM4/IMA subtypes. Next step, Enrichment analysis revealed that the IM1-IMD subtypes were mainly closely related to the production and secretion of IL-8 and TNF signaling pathway. The IM2-IMB subtypes were strongly associated with leukocyte

**Data Availability Statement:** All relevant data are within the paper and its Supporting Information files.

**Funding:** This paper is funded by Anhui provincial key clinical specialties of the 14th Five-Year Plan

(2021-25), and Natural Science Research Project of Colleges and Universities in Anhui Province (No. KJ2021A0293), and Research Funding for Doctoral Talent in 2024 – Feng Yuan (1931). The funders had no role in study design, data collection and analysis, decision to publish, or preparation of the manuscript.

**Competing interests:** The authors have declared that no competing interests exist.

**Abbreviations:** TME, tumor microenvironment; IME, immune microenvironment; ML, Machine Learning; CNS, Central Nervous System; ssGSEA, single sample gene set enrichment analysis; TMZ, temozolomide; GBM, Glioblastoma; TTFields, Tumor-treating fields; BBB, blood-brain barrier; TAMs, Tumor-associated macrophages; Treg, Regulatory T; CTLs, cytotoxic T lymphocytes; NB, Naïve Bayes; SVM, support vector machines; kNN, k-Nearest Neighbors; RF, Random Forest; DNN, Deep Neural Networks; IDH, isocitrate dehydrogenase; CGGA, Chinese Glioma Genome Atlas; TCGA, the Cancer Genome Atlas; DAVID, Database for AnnotationVisualization and Integrated Discovery; GO, Gene ontology; KEGG, Kyoto Encyclopedia of Genes and Genomes; LR, Logistic Regression; DT, Decision Tree; RF, Random Forest; MLP, Multi-Layer Perceptron; LSTM, Long Short-Term Memory; NPV, negative predictive values; DEGs, differentially expressed genes; EMT, epithelial-mesenchymal transition; ISR, immunosilencing radiomic; IAR, immunoactivated radiomic.

activation and NK cell mediated cytotoxicity. The IM3-IMC subtypes were closely related to mitotic nuclear division and mitotic cell cycle process. The IM4-IMA subtypes were strongly associated with Central Nervous System (CNS) development and striated muscle tissue development. Afterwards, Single sample gene set enrichment analysis (ssGSEA) showed that stemness maintenance phenotypes were mainly enriched in the IM4/IMA subtypes; Neuronal phenotypes were closely associated with the IM2/IMB subtypes; and mesenchymal phenotypes and tumorigenic cytokines were highly correlated with the IM2 /IMB subtypes. Finally, we found that compared with patients in the IM2/IMB and IM4/IMA subtypes, the IM1/IMD and IM3/IMC subtypes have the highest proportion of GBM patients, the shortest average overall survival of patients and the lowest proportion of patients with IDH mutation and 1p36/19q13 co-deletion.

## Conclusions

We developed a highly reliable model for predicting different immune subtypes of glioma by ML. Then, we comprehensively analyzed the immune infiltration, molecular and clinical features of different immune subtypes of gliomas and defined gliomas into four subtypes: immunogenic subtype, adaptive immune resistance subtype, mesenchymal subtype, and immune tolerance subtype, which represent different TMEs and different stages of tumor development.

## Introduction

Malignant primary brain tumors are still one of the refractory tumors in humans, its 5-year overall survival rate does not exceed 35% [1]. Glioma is the most common primary central nervous system (CNS) tumor, accounting for about 27% of all CNS tumors and about 80% of intracranial malignant tumors. Among them, Glioblastoma (GBM) has the highest degree of malignancy and the worst prognosis [2]. Although the patient received the standard treatment, that is, the maximum safe removal of the tumor, postoperative radiotherapy, concomitant chemotherapy with temozolomide (TMZ) and Tumor-treating fields (TTFields), the median overall survival is 20.9 months [3,4].

Studies have shown that for patients with GBM who relapse without treatment or after treatment, Surgical removal of more than 98% of the tumor tissue can significantly improve the patient's survival prognosis, while the removal of 70% of the tumor tissue is the lowest resection load that could improve the patient's survival prognosis [5]. Tumor cells in patients with glioma often invade and migrate around early in the brain. The tumor grows diffusely and infiltrates important brain functional areas and deep brain tissues, and spreads along the white matter fiber tracts, adventitial membranes, and unmyelinated axons. Therefore, whether it is a low-grade or high-grade glioma, there is no clearly and distinctly tumor growth boundary. Moreover, functional boundaries are difficult to determine before surgery [6–8].

Why is the prognosis of patients with GBM so poor? Why is the progress of GBM research so slow? In recent years, related studies have found that the reasons may be heterogeneity and tumor evolution, blood-brain barrier (BBB), radiotherapy and chemotherapy resistance, immunosuppressive microenvironment and so on [9,10].

In recent years, the biggest progress in the field of tumor therapy is tumor immunotherapy. More and more new drugs are used to activate the body's own immune system to attack

tumors, which has become the focus of tumor treatments. For example, different forms of immunotherapy include checkpoint blockers, cancer vaccines, CAR T cells and oncolytic virus therapy [11]. The limitations of traditional therapy include: 1.Mainly targeted to kill tumor cells with strong proliferation ability 2.Produce drug resistance 3.Strong side effects; compared with traditional treatment methods, tumor immunotherapy has the following advantages: 1.Mainly rely on the autoimmune system. 2.Adapt to the evolution of tumors. 3. Fewer side effects. 4.Immune memory function. 5.Combined with conventional treatments [12].

However, current immunotherapy could only help a small number of patients with specific cancer types, including bladder cancer, head and neck cancer, kidney cancer, liver cancer, melanoma and non-small cell lung cancer. In some types of cancer, there is almost no response, such as GBM, ovarian cancer, prostate cancer, and pancreatic cancer [13,14]. Therefore, the current research focus is to understand the mechanism by which immunotherapy works or fails, and how to improve it to achieve the desired effect, making it truly a revolutionary treatment for cancer.

Current studies have found that the tumor microenvironment, especially the immune microenvironment of patients with GBM, is in an extremely immunosuppressive state [15]. The following reasons jointly promote the formation of the immunosuppressive microenvironment of GBM: 1.Tumor cells can increase the expression of immunosuppressive-related genes, reduce the expression of MHC and limit the presentation of autoantigens;2.Microglia could secrete TGF-β and IL-10, down-regulate local bone marrow and lymphatic immune cells, and promote systemic immune suppression;3.Tumor-associated macrophages (TAMs) derived from bone marrow cells can suppress immune response and promote tumor progression;4.Regulatory T (Treg) cells could up-regulate various soluble cytokines, immune checkpoint molecules and metabolic pathways to mediate immune suppression;5. The cytotoxic T lymphocytes (CTL) are in a state of exhaustion and highly express exhaustion markers [16,17].

Machine learning (ML) is an emerging and a growing field in the realm of artificial intelligence (AI). It uses different statistical techniques to enable computers to learn from various data types without being explicit. Many classification and clustering solutions in biology, medicine, precise phenotyping and clinical diagnosis support systems utilize ML methods. ML includes many methods, such as Naïve Bayes (NB), support vector machines (SVM), k-Nearest Neighbors (kNN), Random Forest (RF) and Deep Neural Networks (DNN) [18]. Part of these methods are the "unsupervised learning" technology, which can be used to model and learn multiple omics data. Generally, this type of ML method attempts to identify meaningful inferences from data sets that lack classification and classification labels [19]. For example, due to the high development of computer science and medical image, ML in the field of medical image analysis is of great significance to promote the further development of brain tumor imaging diagnosis. Specifically, the radiology-based ML method can accurately predict the isocitrate dehydrogenase (IDH) genotype and 1p/19q coding status of glioma [20]. ML technology based on MRI radiation, genetics and clinical risk factors improves the survival prediction of advanced glioma [21]. Therefore, ML has important significance and promising application prospects for the classification and precise diagnosis of glioma.

In this study, we developed a ML model based on a new classification of immune infiltrating in gliomas, and then comprehensively analyzed the molecular and clinical characteristics of different immune subtypes of gliomas. Our purpose is to reveal some truth about the unsatisfactory results of glioma immunotherapy and to provide guidance for glioma immunotherapy.

## Materials and methods

### Databases and samples

The two databases utilized in this study are sourced from The Cancer Genome Atlas (TCGA) and the Chinese Glioma Genome Atlas (CGGA). The mRNA sequencing data from TCGA, specifically the Illumina HiSeq platform (n = 702), was downloaded from UCSC Xena (https://xena.ucsc.edu/), while the corresponding clinical and molecular pathological data were obtained from the TCGA website (https://tcga-data.nci.nih.gov/docs/publications/lgggbm_2015/). The mRNA sequencing data and clinical information were acquired from CGGA database (http://www.cgcg.org.cn/, n = 693). This study adopts the Reads Per Kilobase of exon model per Million mapped reads (RPKM) method for the normalization of mRNA sequencing data.

### Single sample gene set enrichment analysis

Single sample gene set enrichment analysis (ssGSEA) normalizes the gene expression profile within a sample and then calculates the ssGSEA score for each gene set. In this way, ssGSEA transforms the gene expression profile of a single sample into a gene set enrichment score matrix.

### Identification of Differentially Expressed Genes (DEGs)

Morpheus is a multifunctional matrix visualization and analysis software used for screening DEGs across different subtypes of glioma. A P-value of <0.01 and a False Discovery Rate (FDR) of <0.01 are considered critical thresholds.

### Gene ontology (GO) and Kyoto Encyclopedia of Genes and Genomes (KEGG) pathway analysis

The online tool, Database for Annotation, Visualization, and Integrated Discovery (DAVID, https://david.ncifcrf.gov/), provides a comprehensive biological functional annotation for gene or protein lists through GO and KEGG pathway analysis. Additionally, GO enrichment analysis encompasses three distinct categories: Biological Process (BP), Molecular Function (MF), and Cellular Component (CC). DAVID assists in the interpretation of genome-scale datasets by facilitating the transition from data collection to biological meaning [22]. A P-value of < 0.05 is considered statistically significant.

### Machine-Learning (ML) methods for different immune subtypes classification

Predictive models were developed using representative supervised ML algorithms, including Logistic Regression (LR), Decision Tree (DT), Random Forest (RF), Multi-Layer Perceptron (MLP) and Long Short-Term Memory (LSTM). ROC-analysis and performance characteristics were applied to assess the accuracy of models, including sensitivity, specificity, positive and negative predictive values (PPV and NPV). Based on our sample size, 7-fold cross-validation was applied for each iteration, and 600 patients were used to train the models during each iteration. A total of 122 immune related metagene clusters and dozens of DEGs (differentially expressed genes) of each immune subtypes were used separately to train and test our models. All models were developed and evaluated in Python version 3.7 (Python Software Foundation, Delaware, United).

### Statistical analysis

In this study, both databases were standardized, and the data were analyzed using Graph Pad Prism 6.0, SPSS19.0, R3.3.2 statistical software and Python Software. One-way analysis of variance was used for the differences between the three groups. The analysis of differences between the two groups used unpaired t test. P<0.05 indicates that the difference is statistically significant. Kaplan-Meier survival curve was used to analyze the survival difference of patients with different immune subtypes.

## Results

### Glioma classification based on immune infiltration

In order to explore the characteristics of the immune microenvironment in gliomas, firstly, we selected 122 immune related metagene clusters (S1 Table), which represent eight different types of immune cells (including B lineage, CD8 T cells, Cytotoxic lymphocytes, Monocytic lineage, Myeloid dendritic cells, Neutrophils, NK cells and T cells). Then, we performed unsupervised cluster analysis on the metagene clusters of different immune cells and patient arrays in 693 glioma patients from CGGA database and 702 glioma patients from TCGA database (Fig 1A and 1B). The results found that the immune microenvironment of glioma patients in the CGGA database can be divided into four different subtypes, namely IM1, IM2, IM3, and IM4; similarly, glioma patients in the TCGA database can also be divided into four different subtypes (IMA, IMB, IMC, and IMD). Finally, the Kaplan-Meier survival curve was used to evaluate the prognostic characteristics of glioma patients in four different immune subtypes (Fig 1C and 1D). The results found that the IM1 group in the CGGA database and the IMD group in the TCGA database had the worst prognosis. Both the IM2 group in the CGGA database and the IMB group in the TCGA database have the best prognosis. Based on the above results, we conclude that based on the different immune microenvironment types of gliomas, the new immune classification of gliomas can be divided into four immune subtypes.

### Establishment and verification of different immune subtypes prediction models of glioma based on machine learning (ML)

In order to improve the credibility of the new immune classification of glioma and popularize the application of the new immune classification, we developed prediction models for different immune subtypes of glioma based on ML, then two methods of 122 immune related metagene clusters and dozens of DEGs (P <0.01 and FDR <0.01) (S2 Table) of each immune subtypes were used to train and test the prediction models (Fig 2A, 2B and 2D). Finally, it was found that the prediction model has high accuracy in predicting different immune subtypes of gliomas (Fig 2C).

### Distribution, expression and prognostic characteristics of different types of immune cells in different immune subtypes of glioma

In order to explore the distribution, expression and prognostic characteristics of different types of immune cells in different immune subtypes of glioma (S3 Table); Kaplan-Meier survival curve analysis and cluster analysis showed that B lineage, Myeloid dendritic cells, Neutrophils and CD8 T cells were used as prognostic risk factors in the CGGA database, while Cytotoxic lymphocytes and Monocytic lineage were used as prognostic protective factors (Fig 3C). Interestingly, in the TCGA database, B lineage, Myeloid dendritic cells, Neutrophils and Monocytic lineage are used as prognostic risk factors, while Cytotoxic lymphocytes are used as prognostic protective factors (Fig 3D). At the same time, it was also found that B

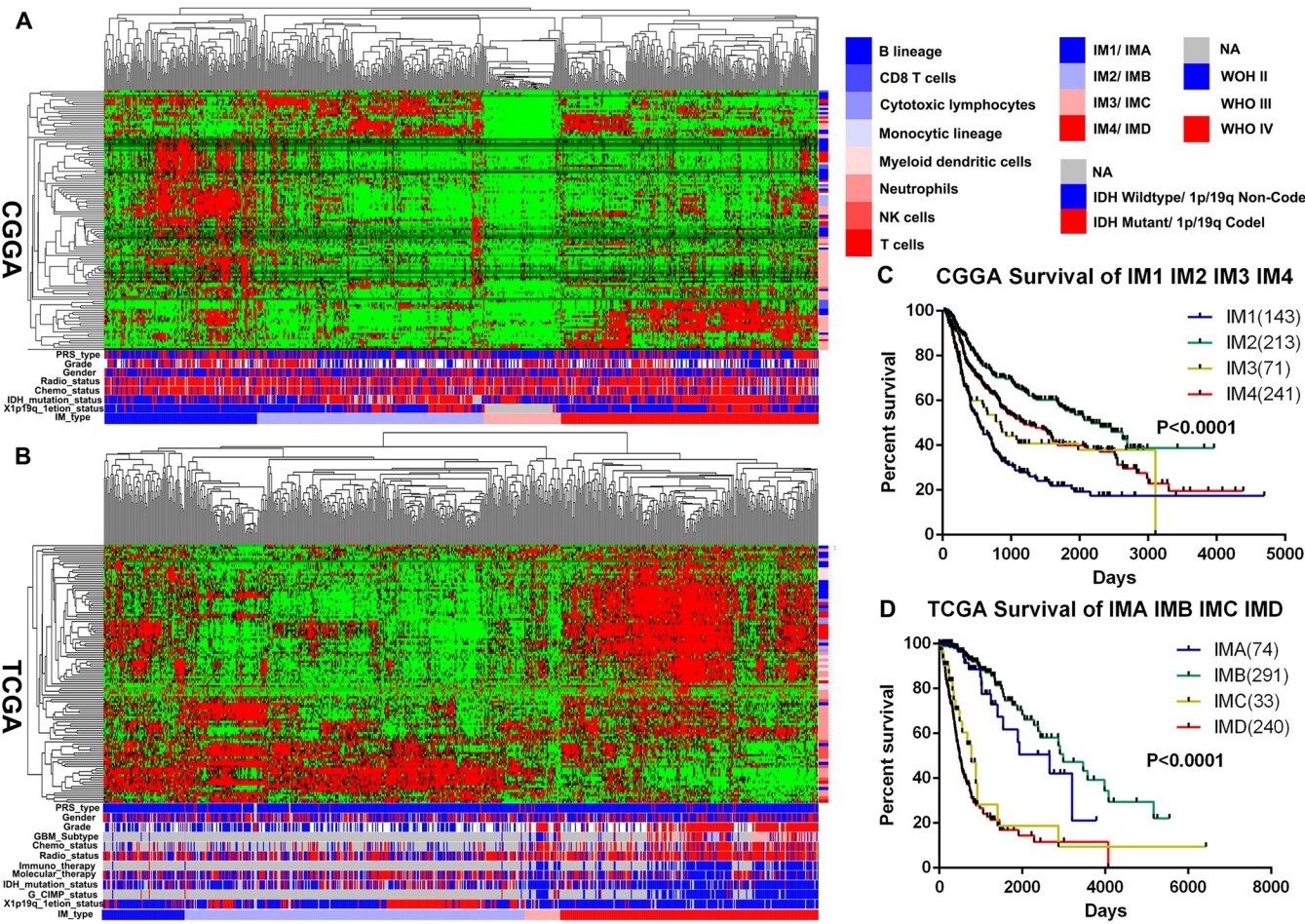

**Fig 1. Glioma classification based on immune infiltration.** (A, B) Through unsupervised cluster analysis of the molecular markers of different immune cells and patient arrays in 693 glioma patients from CGGA database and 702 glioma patients from TCGA database, the immune microenvironment of gliomas can be divided into four different subtypes, namely IM1/IMA, IM2/IMB, IM3/IMC, IM4/IMD. (C, D) The Kaplan-Meier survival curve was used to evaluate the prognostic characteristics of glioma patients in four different immune subtypes. *, p<0.05; **, p<0.01; ***, p<0.001; ****, p<0.0001; NS, not significant.

lineage, Monocytic lineage, Myeloid dendritic cells, Neutrophils, NK cells, T cells and CD8 T cells had the highest expression or enrichment in the IM1 group of the CGGA database and the IMD group in the TCGA database. Cytotoxic lymphocytes expressed or enriched the highest in IM4 group of CGGA database and IMA group of TCGA database (Fig 3A–3D). In summary, the immune cells in different immune subtypes of glioma have obvious differences in distribution, and they play an important role in the development and prognosis of gliomas.

## GO enrichment analysis and KEGG pathway analysis in different immune subtypes of glioma

In order to further explore the molecular mechanisms and signaling pathways in different immune subtypes of glioma, we first divided the immune subtypes in the CGGA database and TCGA database into IM1-IMD, IM2-IMB, IM3-IMC and IM4-IMA based on molecular and clinical prognostic characteristics.

Then select the DEGs (P <0.01 and FDR <0.01) of each matched immune subtypes for GO enrichment analysis and KEGG pathway analysis (The heatmap displays the top 100 DEGs)

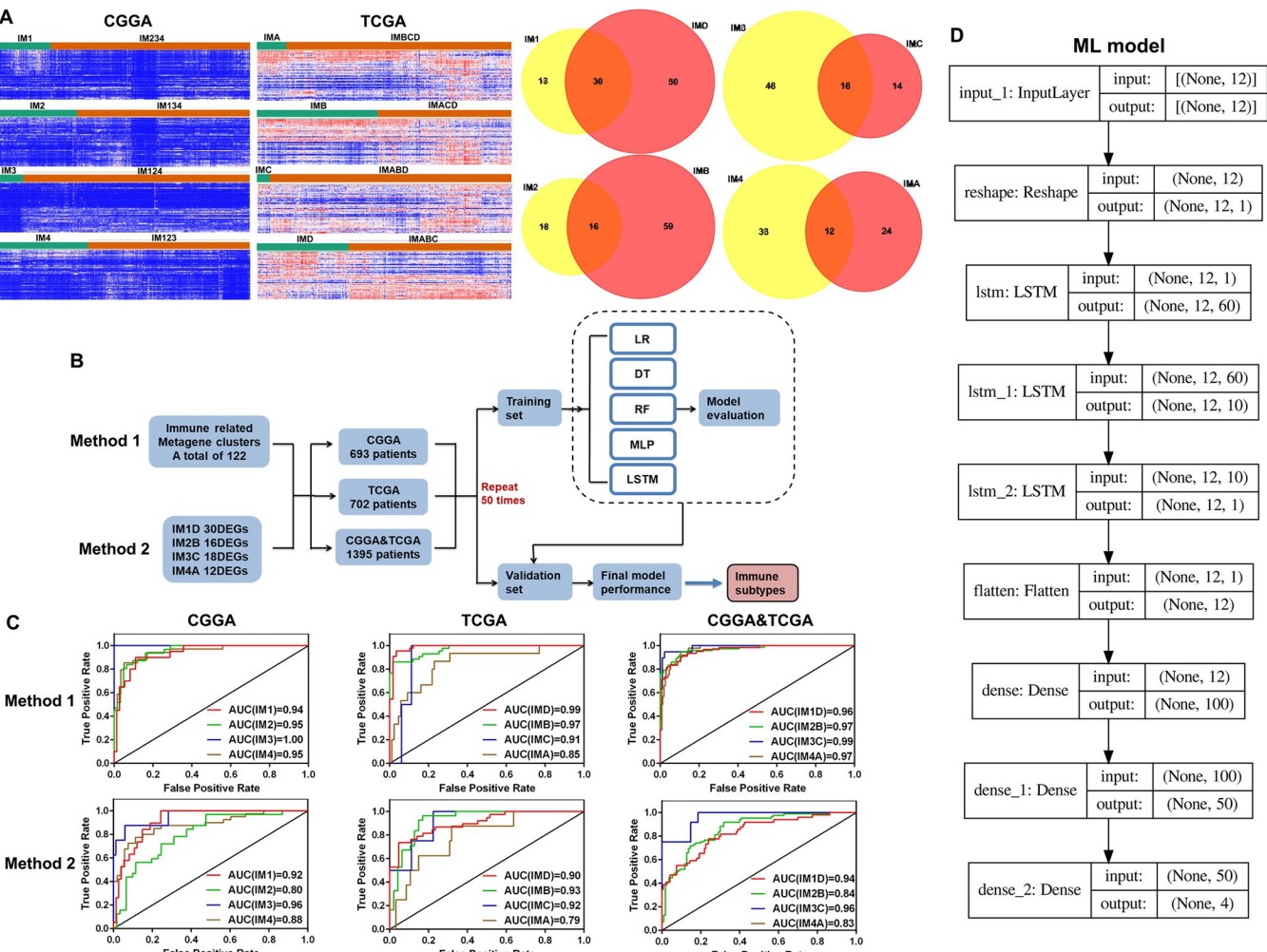

**Fig 2. Establishment and verification of different immune subtypes prediction models of glioma based on machine learning (ML).** (A) The heat map and Venn map show the DEGs of each immune subtype. (B, D) The flowchart shows the ML process of prediction models for different immune subtypes of glioma. (C) The ROC curve of each immune subtypes prediction models.

(Fig 4A). The results found that the IM1-IMD group is mainly closely related to the production and secretion of IL-8, TNF signaling pathway and NF-kappa B signaling. The IM2-IMB group is strongly associated with leukocyte activation, MAPK signaling pathway and NK cell mediated cytotoxicity. The IM3-IMC group is closely related to mitotic nuclear division and mitotic cell cycle process. The IM4-IMA group is strongly associated with CNS development and striated muscle tissue development (Fig 4B and 4C). The above results reveal that different immune subtypes of gliomas represent different tumor-related biological processes and different stages of tumor development.

## Characteristics of immunosuppression in different immune subtypes of glioma

In order to explore the immunosuppressive characteristics of different immune subgroups of glioma, we selected immune checkpoint molecules, tumor-associated macrophages and Treg cell-related molecular markers for evaluation (S4 Table). The results found that immune

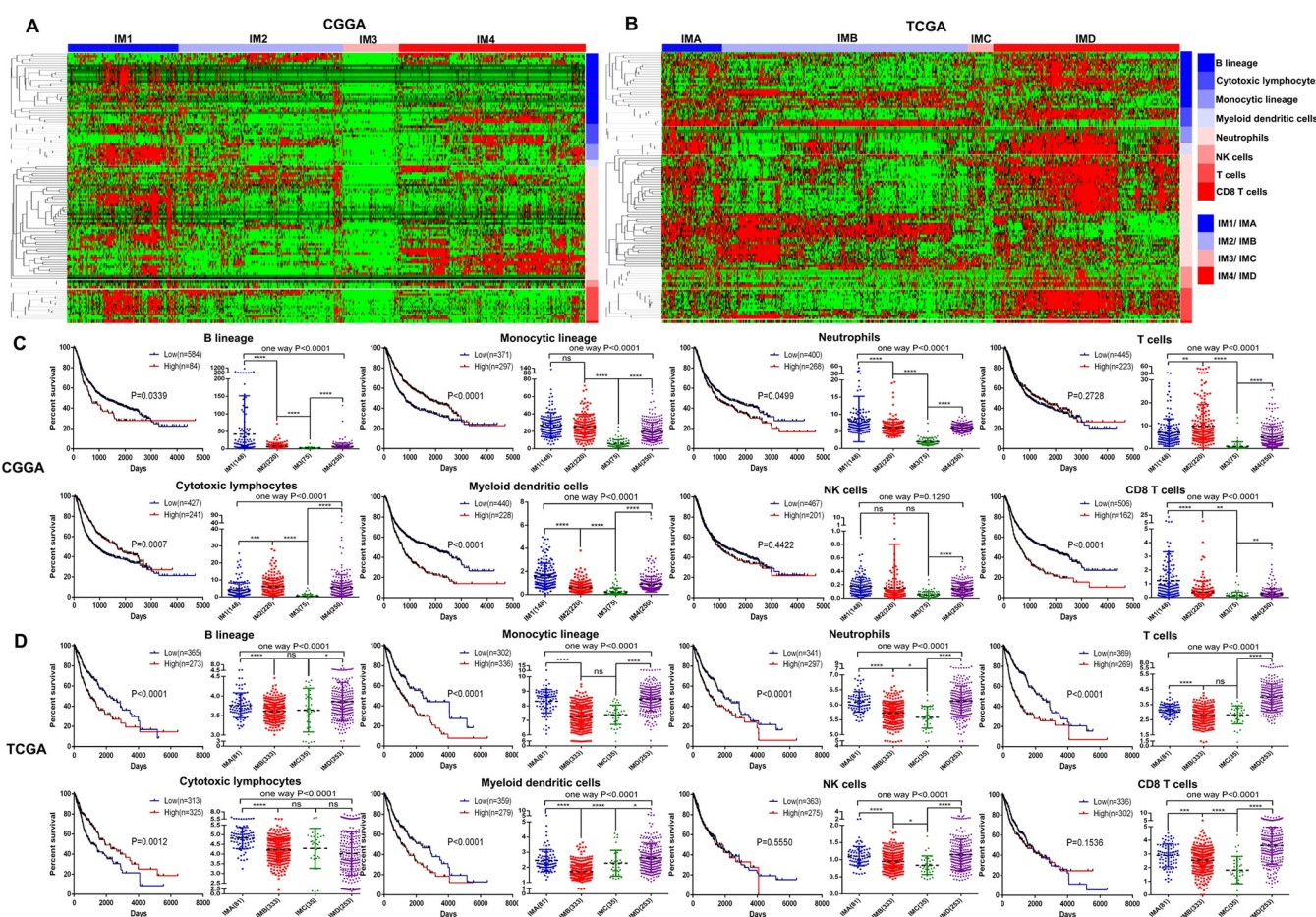

**Fig 3. Distribution, expression and prognostic characteristics of different types of immune cells in different immune subtypes of glioma.** (A, B) The heat map showed the results of unsupervised cluster analysis of different immune cells in the CGGA and TCGA databases. (C, D) The Kaplan-Meier survival curve and Scatter plot showed the prognostic characteristics and expression of different immune cells in each immune subtype.

checkpoint molecules, tumor-associated macrophages and Treg cells in the CGGA database and TCGA database are all used as prognostic risk factors for glioma (Fig 5C and 5D); It was also found that immune checkpoint molecules, tumor-associated macrophages and Treg cell-related molecular markers had the highest expression or enrichment in the IM1 subtype of the CGGA database and the IMD subtype of the TCGA database (Fig 5A and 5B). The above results indicate that the tumor immune microenvironment has the highest immunosuppressive intensity in the IM1 subtype of the CGGA database and the IMD subtype of the TCGA database, which exactly corresponds to the worst prognosis of the two subtypes of glioma patients.

## Characteristics of stemness maintain, mesenchymal phenotype, neuronal phenotype and tumorigenetic cytokines distribution in different immune subtypes of glioma

In order to explore the distribution characteristics of stemness maintain, mesenchymal phenotype, neuronal phenotype and tumorigenetic cytokines in different immune subtypes of glioma. We selected stemness maintain, mesenchymal phenotype, neuronal phenotype and tumorigenetic cytokines-related molecular markers for evaluation (S5 Table). The results

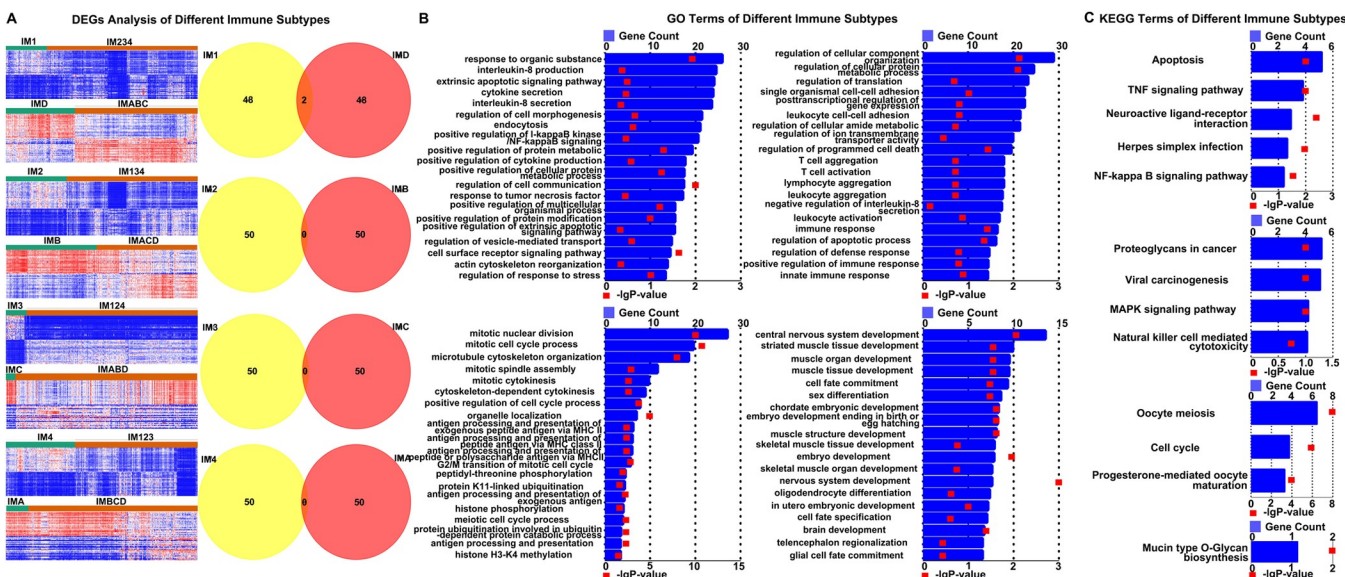

**Fig 4. GO enrichment analysis and KEGG pathway analysis in different immune subtypes of glioma.** (A) The heat map and Venn map show the DEGs (differentially expressed genes) of each immune subtype. (B, C) GO enrichment analysis and KEGG analysis of different matched immune subtypes in CGGA and TCGA databases.

showed that the mesenchymal phenotype and tumorigenetic cytokines-related molecular markers had the highest expression or strongly enrichment in the IM1 subtype of the CGGA database and the IMD subtype of the TCGA database. The molecular markers related to

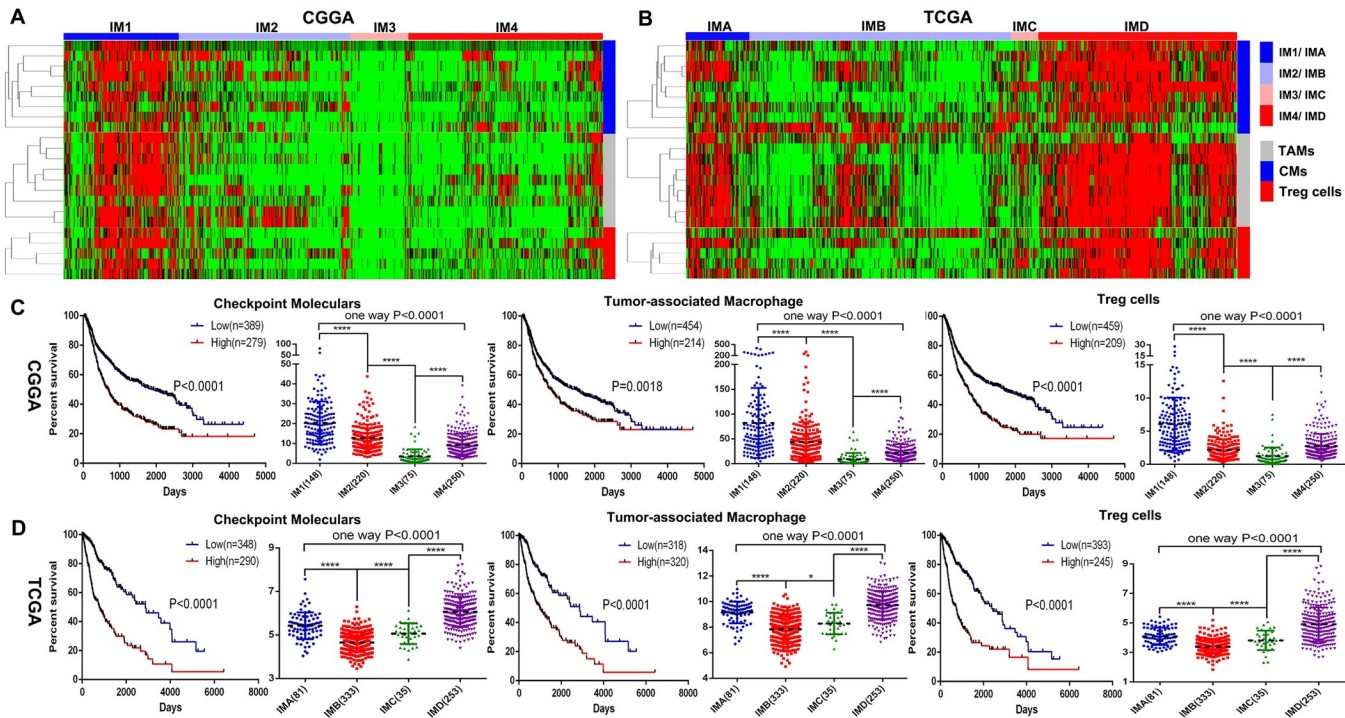

**Fig 5. Characteristics of immunosuppression in different immune subtypes of glioma.** The heat map showed the distribution characteristics of TAMs, CMs (checkpoint molecules) and Treg cells in each immune subtypes. (C, D) The Kaplan-Meier survival curve and Scatter plot showed the prognostic characteristics and expression of TAMs, CMs and Treg cells in each immune subtype.

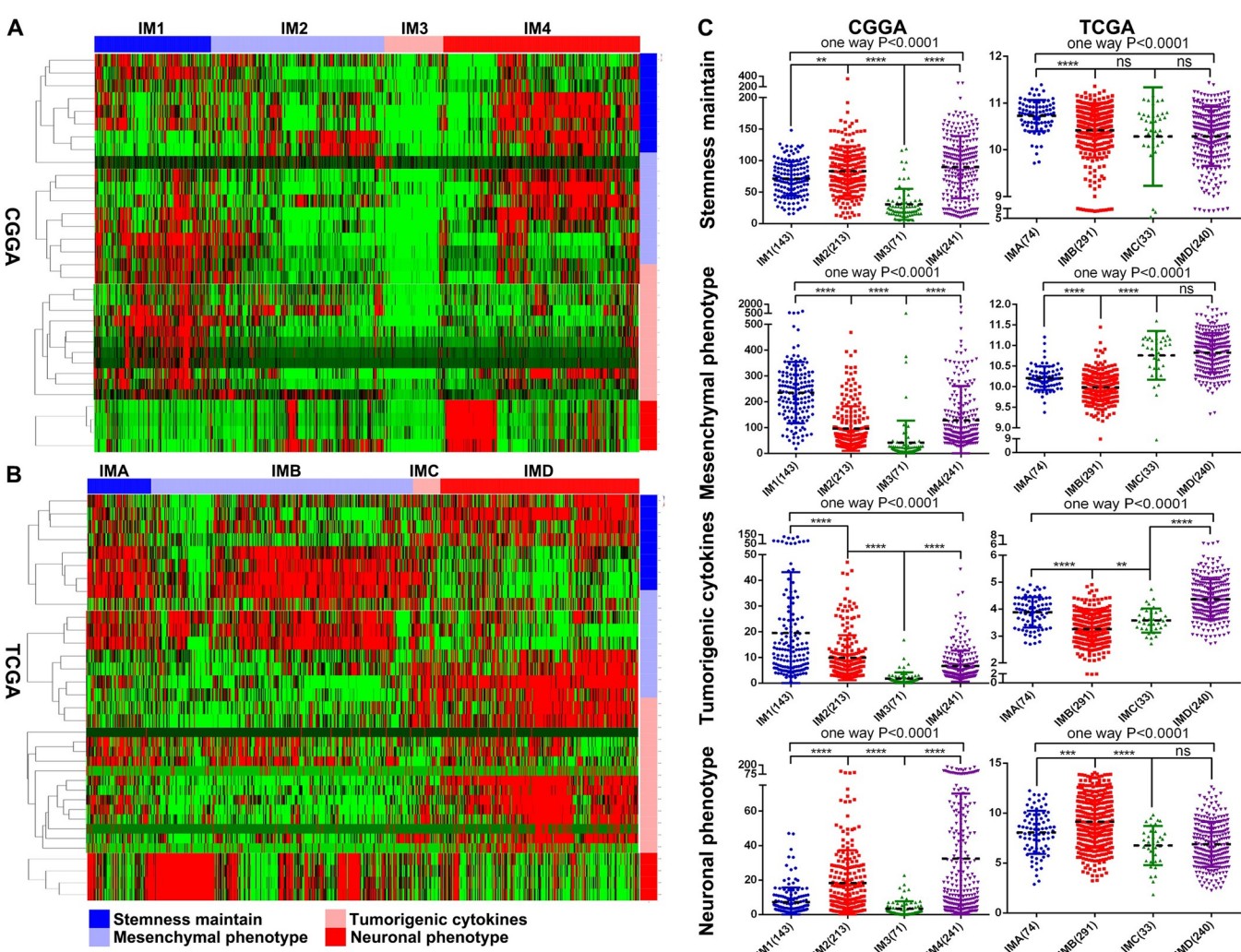

**Fig 6. Characteristics of stemness maintain, mesenchymal phenotype, neuronal phenotype and tumorigenetic cytokines distribution in different immune subtypes of glioma.** (A, B) The heat map showed the distribution characteristics of stemness maintain, mesenchymal phenotype, neuronal phenotype and tumorigenetic cytokines in each immune subtypes. (C)The scatter plot showed the the expression characteristics of stemness maintain, mesenchymal phenotype, neuronal phenotype and tumorigenetic cytokines-related molecular markers in each immune subtype. ns, *, ** and *** indicate p < 0.05, p < 0.01 and p < 0.0001, respectively.

stemness maintain have the highest expression or strongly enrichment in the IM4 subtype of CGGA database and the IMA subtype of TCGA database. The neuronal phenotype-related molecular markers have the highest expression or strongly enrichment in the IM2 group of CGGA database and the IMB group of TCGA database (Fig 6A–6C). The above results also indicate that different immune subtypes of gliomas represent different tumor microenvironments and different stages of tumor development.

## Molecular and clinical features in different immune subtypes of glioma

In order to further explore the molecular and clinical characteristics of different immune subtypes of glioma, we selected IDH mutation, 1p36/19q13 co-deletion, age, gender, WHO pathological grade, sample type, overall survival and other molecular and clinical indicators for analysis. The results found that compared with patients in other immune subtype groups, the IM1 subtype of CGGA database and the IMD subtype of TCGA database have the highest

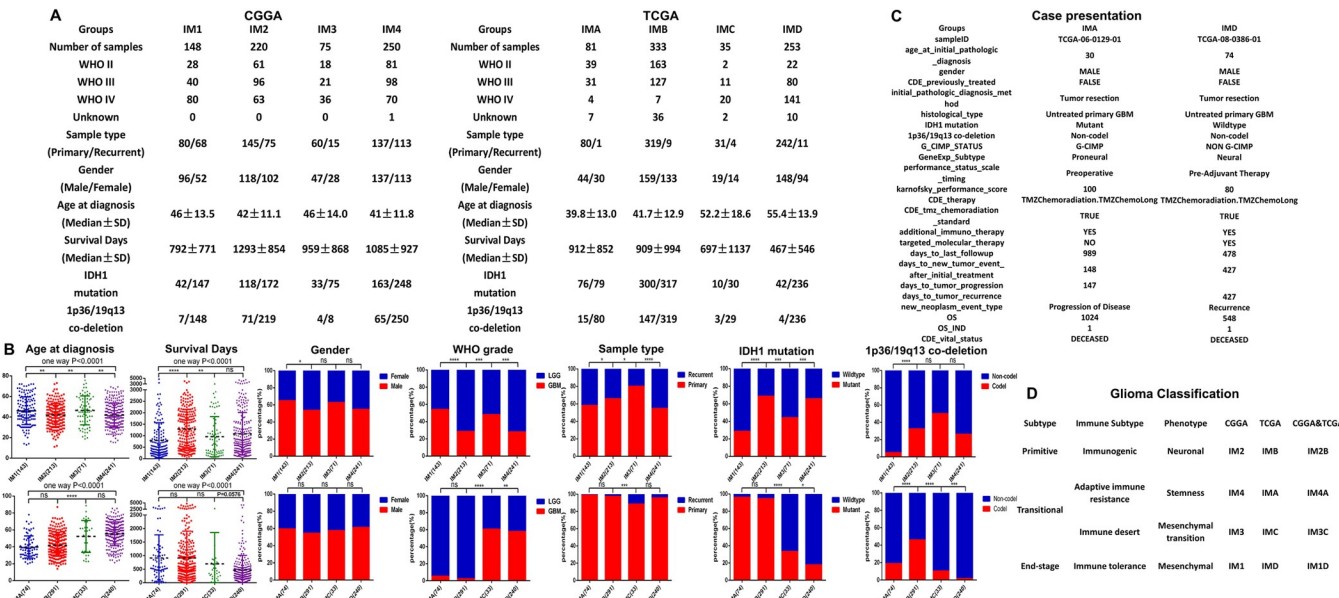

**Fig 7. Molecular and clinical features in different immune subtypes of glioma.** (A, B) Summary and statistical analysis of the molecular and clinical characteristics of different immune subtypes in CGGA and TCGA databases. (C) The molecular and clinical characteristics of two patients with primary GBM who have received standard treatment and immunotherapy in the TCGA database. (D) New classification and naming of different immune subtypes of gliomas.

proportion of WHO IV patients, the shortest average overall survival of patients and the lowest proportion of patients with IDH mutation and 1p36/19q13 co-deletion. Interestingly, the IM2 subtype of CGGA database and the IMB subtype of TCGA database have the highest proportion of Lower Grade Glioma (LGG) patients, the longest average overall survival of patients and the highest proportion of patients with IDH mutation and 1p36/19q13 co-deletion (Fig 7A and 7B). We analyzed the molecular and clinical information of two patients with primary GBM without any preoperative treatment in the TCGA database. Both patients underwent standard Stupp protocol (the maximum safe removal of the tumor, postoperative radiotherapy and concomitant chemotherapy with TMZ) and all received immunotherapy. Case 1 belongs to the IMA subtype, with an overall survival period of 1024 days, and tumor progression occurred 147 days after treatment; Case 2 belongs to the IMD subtype with an overall survival period of 548 days, and tumor recurrence occurred 427 days after treatment (Fig 7C). We analyzed the molecular and clinical characteristics of two patients with primary GBM who received standard treatment and immunotherapy, and found that they were fully consistent with the new classification of glioma in our study. Based on the molecular and clinical characteristics of the above-mentioned different immune subtypes of gliomas (two of the patients received immunotherapy), we inferred that the different immune subtypes of gliomas represent different tumor microenvironments and different stages of tumor occurrence and development. The treatment response of related therapies in different immune subtypes of glioma is different. Based on the molecular and clinical characteristics of different immune subtypes of glioma, we have made new classifications and names for different immune subtypes of glioma (Fig 7D).

## Discussion

Glioma is the most common primary CNS malignancy, which is a fatal and poorly prognostic tumor [23]. Immunotherapy has achieved unprecedented success in certain advanced cancer

patients and has prolonged the lives of cancer patients [24]. Malignant glioma is a malignant tumor with poor immunogenicity. So far, the latest immunotherapy has not made break-through progress in patients with glioma [25]. Therefore, understanding the mechanism by which immunotherapy works or fails, and how to improve it to achieve the desired effect has become the focus of current research.

Recent studies have shown that the brain is also a tissue with an immune system. But the immune level of the brain is changing dynamically. The first line of defense in the brain's defense system is the BBB and the microglia in the tissues. In the resting state, peripheral immune cells are excluded from the CNS. In an inflammatory state, cytokines induced by interferon in the peripheral immune system pass through the BBB and enter the brain [26,27]. Both the immune cells inside the brain and the peripheral immune cells closely monitor the antigens in the brain. Once a danger signal appears, peripheral immune cells will cross the BBB and produce an inflammatory response. These also provide the basis for immunotherapy of brain tumors [28].

The model "Three Es Hypothesis" proposed in 2002 summarizes the resistance of tumor cells and immune cells in treatment. Both the intrinsic resistance and adaptive resistance of tumor cells are involved in this process [29]. Gliomas, especially GBM have a high degree of endogenous resistance mechanism and exogenous acquired resistance mechanism; Tumor cells can evade external immune pressure through a variety of mechanisms such as inducing immune response barriers and acquired immunotherapy resistance [30,31].

Gliomas, especially GBM tumors, are highly heterogeneous. Immunotherapy kills the treatment-sensitive cells, but the treatment-resistant cells will grow rapidly. Molecular heterogeneity is also an important mechanism of endogenous resistance in GBM. Intratumoral heterogeneity is one of the main obstacles to the effectiveness of GBM immunotherapy [32,33]. A recent study found that short peptide vaccines are safe and effective for 82% of relapsed GBM that do not express EGFR triisomers. However, subsequent clinical trials found that the victory of early immunotherapy indicates that later tumors will evade treatment, and the clinical trial was forced to terminate [10,34].

Recent studies have found that there are a large number of myeloid-derived cells in the tumor tissues of GBM patients [35]. The tumor cells recruit TAMs and promote their polarization toward the anti-inflammatory M2 TAMs type [36]. TAMs can promote the occurrence and development of tumors through a variety of ways, including promoting genomic instability, supporting tumor stem cell growth, promoting epithelial-mesenchymal transition (EMT) and up-regulating the expression of immune checkpoint ligands to inhibit anti-tumor adaptive immune responses [37,38].

The application of immunotherapy to clinical tumors has greatly improved the treatment efficiency of highly malignant tumors [39]. In glioma patients, the therapeutic effects of various immunotherapies are still very unsatisfactory [40]. Based on the current status of glioma immunotherapy, the characteristics of the immune microenvironment of glioma and the results of this study, we divide gliomas into four different immune subtypes, namely immunogenic subtype, adaptive immune resistance subtype, and immune desert subtype and immune tolerance subtype, Different immune subtypes represent different tumor microenvironments and different stages of tumor development. Patients with immunogenic subtype have better immunotherapy effects, while those with immune tolerance subtype have poor immunotherapy effects. Sun Y et al. utilized imaging data from 200 gliomas and employed a Mask R-CNN-based deep learning approach to classify these patients into two subtypes. One subtype, defined as the immunosilencing radiomic (ISR) subtype, exhibited a poorer prognosis, while the other subtype was identified as the immunoactivated radiomic (IAR) subtype. The study compared the immune infiltration patterns, genomic characteristics, specific drug responses, and

predictive models between these subtypes. This finding provides new insights for the precision treatment of gliomas [41]. Hoogstrate et al. compared the transcriptomes of primary and recurrent IDH-wildtype GBMs, revealing that transcriptional subtypes form interconnected continua in two-dimensional space, with recurrent tumors displaying a predominant mesenchymal progression. Over time, hallmark glioblastoma genes do not show significant changes. In contrast, tumor purity decreases over time, accompanied by a concurrent increase in neuronal and oligodendrocytic marker genes as well as independent TAMs. Their data suggest that GBM mainly undergoes microenvironmental reorganization rather than molecular evolution of tumor cells. A better understanding of the interplay between tumor cells, their microenvironment, and the effects of chemotherapy and radiotherapy may be a crucial direction for improving GBM treatment [42].

We infer that the four immune subtypes of glioma are likely to be a dynamic process. From the immunogenic subtype in the primitive state to the adaptive immune resistance subtype and immune desert subtype in the transitional state, and finally to the immune tolerance subtype in the end-stage state. According to the immune status and characteristics of patients with each immune subtype, immunotherapy strategies for different immune subtypes should be different. For example, patients with immunogenic subtype of glioma in the primitive state should mainly actively mobilize immune cells and increase the intensity of immune response. While patients with immune tolerance subtype in the end-stage state should mainly actively suppress tumor-related immunosuppressive cells. At the same time, we also proposed that immunotherapy should be used as the first-line treatment for glioma patients, and should not be used until glioma patients are in the terminal stage.

In our current research, we use ML methods to establish and verify different immune subtype models of glioma based on immune infiltration, then comprehensive analysis of the molecular and clinical characteristics of different immune subtypes of glioma. We uncovered that different immune subtypes of gliomas represent different tumor microenvironments and different stages of tumor development. The therapeutic response of immune-related therapies in different immune subtypes of glioma is different. This study revealed part of the truth about the unsatisfactory results of glioma immunotherapy, and provided guidance for glioma immunotherapy.

The primary limitation of this study is that the immune subtype classification derived solely from transcriptomic data using ML methods lacks support from single-cell sequencing data and clinical validation. In the next phase, we will incorporate single-cell sequencing data from clinical gliomas at our center, as well as relevant clinical immunotherapy data, to further validate and refine the conclusions of this study.

## Conclusion

In summary, this study developed a highly reliable model for predicting different immune subtypes of glioma by ML. Then, we comprehensively analyzed the immune infiltration, molecular and clinical features of different immune subtypes of gliomas and defined gliomas into four subtypes: immunogenic subtype, adaptive immune resistance subtype, mesenchymal subtype, and immune tolerance subtype, which represent different TMEs and different stages of tumor development.

## Supporting information

**S1 Table. Immune related metagene clusters in TCGA and CGGA databases.**
(XLSX)

**S2 Table. DEGs of different immune subtype in TCGA and CGGA databases.**
(XLSX)

**S3 Table. Characterization of immune infiltration of different immune subtypes in the TCGA and CGGA databases.**
(XLSX)

**S4 Table. Immunosuppressive characteristics of different immune subtypes in the TCGA and CGGA databases.**
(XLSX)

**S5 Table. Characterization of functional phenotypes of different immune subtypes in the TCGA and CGGA databases.**
(XLSX)

## Author Contributions

**Conceptualization:** Feng Yuan.

**Data curation:** Feng Yuan, Yingshuai Wang, Lei Yuan, Lei Ye.

**Formal analysis:** Feng Yuan, Yingshuai Wang, Lei Yuan, Lei Ye.

**Funding acquisition:** Hongwei Cheng, Yan Li.

**Investigation:** Yingshuai Wang, Lei Yuan, Lei Ye.

**Methodology:** Feng Yuan, Yingshuai Wang, Lei Yuan, Lei Ye.

**Software:** Yingshuai Wang, Lei Yuan, Lei Ye.

**Supervision:** Feng Yuan, Lei Ye, Yangchun Hu, Hongwei Cheng, Yan Li.

**Validation:** Feng Yuan, Lei Ye, Yangchun Hu, Yan Li.

**Visualization:** Lei Ye, Yan Li.

**Writing – original draft:** Feng Yuan.

**Writing – review & editing:** Yangchun Hu, Hongwei Cheng, Yan Li.

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
