## [Decision Letter · Decision Letter 0]

22 Aug 2024

PONE-D-24-22148Machine Learning–Based New Classification for Immune Infiltration of GliomasPLOS ONE

Dear Dr. Yuan,

Thank you for submitting your manuscript to PLOS ONE. After careful consideration, we feel that it has merit but does not fully meet PLOS ONE’s publication criteria as it currently stands. Therefore, we invite you to submit a revised version of the manuscript that addresses the points raised during the review process.

 Please address the issues brought forward by the reviewers, and modify your manuscript accordingly. Please discuss, where this might not be possible.

We look forward to receiving your revised manuscript.

Kind regards,

Michael C Burger, M.D.

Academic Editor

PLOS ONE

Journal Requirements:

"This paper is funded by Anhui provincial key clinical specialties of the 14th Five-Year Plan (2021-25)."

"We thank the authors of datasets (TCGA, and CGGA) for sharing these valuable data. We special thanks to the support from Anhui provincial key clinical specialties of the 14th Five-Year Plan (2021-25)."

"This paper is funded by Anhui provincial key clinical specialties of the 14th Five-Year Plan (2021-25)."

Reviewers' comments:

Reviewer's Responses to Questions

**Comments to the Author**

1. Is the manuscript technically sound, and do the data support the conclusions?

Reviewer #1: Partly

Reviewer #2: Yes

2. Has the statistical analysis been performed appropriately and rigorously? 

Reviewer #1: Yes

Reviewer #2: Yes

3. Have the authors made all data underlying the findings in their manuscript fully available?

Reviewer #1: No

Reviewer #2: Yes

4. Is the manuscript presented in an intelligible fashion and written in standard English?

Reviewer #1: Yes

Reviewer #2: Yes

5. Review Comments to the Author

Reviewer #1: The authors examined the immunosuppressive tumor microenvironment characteristics in gliomas using data from the TCGA and CGGA databases, employing machine learning unsupervised cluster analysis and DAVID methods. They classified gliomas into subtypes IM1-IM4 and IMA-IMD through machine learning, revealing distinct TME characteristics including immune cell infiltration proportions, IL-8 and TNF signaling pathways, and various biological processes. However, fundamental flaws in the study are evident:

- Subtypes were ambiguously defined.

- Enrichment analysis linked IM1-IMD subtypes with IL-8 and TNF pathways, IM2-IMB with leukocyte activation and NK cell cytotoxicity, IM3-IMC with mitotic processes, and IM4-IMA with CNS and muscle development. These associations are descriptive and lack unified biological significance, let alone clinical relevance.

- Clustering analysis from figures demonstrates unclear differences among groups, and survival analyses lack reproducibility.

Overall, these limitations suggest the manuscript may not meet the threshold for publication priority in the journal.

Reviewer #2: Summary of the research and the overall impression:

The study’s ultimate goal is to discover precise reasons why glioma treatment fails - researchers investigate the immune microenvironment (IME) of gliomas and develop a machine learning (ML) model to classify these tumors based on their immune infiltration characteristics.

The primary research question is to understand the immune microenvironment of gliomas and predict different immune subtypes using ML methods. This aims to guide personalized immune therapy in glioma patients, potentially improving its effectiveness.

The authors employed a combination of bioinformatics tools, statistical analysis, and ML techniques to classify gliomas based on their immune infiltration characteristics. Unsupervised cluster analysis on immune-related metagene clusters derived 4 immune subtypes of glioma in both datasets. ML algorithms used: logistic regression, decision tree, random forest, MLP, LSTM. Gene Ontology (GO) and Kyoto Encyclopedia of Genes and Genomes (KEGG) pathway analysis was performed using DAVID, to understand the biological functions and pathways associated with the different immune subtypes. The distribution of immune cells and their prognostic significance were analyzed using Kaplan-Meier survival curves. The study successfully classified gliomas into 4 distinct immune subtypes using ML: immunogenic subtype, adaptive immune resistance subtype, mesenchymal subtype, and immune tolerance subtype. The immune microenvironment of gliomas shows distinct infiltration patterns of immune cells, with specific subtypes exhibiting high enrichment of certain cell types like Monocytic lineage, Myeloid dendritic cells, NK cells, and CD8 T cells in IM1/IMD subtypes.

A well-written, biologically and clinically relevant discussion includes at the end the recommended treatment strategies for patients with each immune subtype.

The authors have cited relevant literature regarding the challenges in treating gliomas, the role of the immune microenvironment, and the application of machine learning in oncology. This provides a solid reference foundation for their research.

The dataset is obtained from publicly accessible repositories - CGGA and TCGA, encompassing a large sample size of glioma cases.

The authors employ appropriate statistical methods to validate their findings.

The code is not available publicly - it will be shared at the reader’s request.

The authors have followed appropriate protocols for data collection, analysis, and validation.

The manuscript demonstrates a high level of scientific rigor, and relevance to the field of neuro-oncology. It is suitable for publication in its present form, with minor revisions to add clarity and detail.

The manuscript demonstrates conformance to general principles of transparency, data availability, and methodological rigor.

The manuscript is well-organized and easy to follow, making for a pleasant read.

Minor issues:

The manuscript mentions the use of the CGGA and TCGA datasets, but lacks detail on how these datasets were selected, processed, and used in analyses. Please include a detailed description of the datasets, such as the types of data available (e.g., RNA-seq, clinical data), and specific cohorts.

Readers may need more information about the normalization methods for the transcriptomic data.

Please consider describing how the differentially expressed genes (DEGs) were selected, including the criteria for differential expression such as statistical thresholds (e.g., p-value, fold change) and the tools used.

The manuscript briefly mentions using DAVID for GO and KEGG pathway analyses without details on the processes involved. Please consider mentioning more of the DAVID tool overview and specific settings applied.

In the part of the discussion related to proposed strategies for different immune subtypes, it would be beneficial to directly refer to the relevant medical literature to stronger justify the treatment recommendations.

Please acknowledge the study limitations and suggest future research directions.

6. PLOS authors have the option to publish the peer review history of their article (what does this mean?). If published, this will include your full peer review and any attached files.

Reviewer #1: No

Reviewer #2: No

---

## [Author Response · Author response to Decision Letter 0]

27 Sep 2024

Dear Dr. Michael C Burger,

Thank you very much for your time involved in reviewing this manuscript. We have studied your comments carefully and carefully revised the manuscript as suggested.

Q1. Please ensure that your manuscript meets PLOS ONE's style requirements, including those for file naming. The PLOS ONE style templates can be found at https:// journals. plos.org/ plosone/s/file?id=wjVg/PLOSOne_formatting_sample_main_body.pdf and https://journals. plos.org/plosone/s/file?id=ba62/PLOSOne_formatting_sample_title_authors_affiliations.pdf.

Response 1: Thanks for these suggestions. We referenced the PLOS ONE style template and made the necessary modifications to meet its requirements.

Q2. Please state what role the funders took in the study. If the funders had no role, please state: "The funders had no role in study design, data collection and analysis, decision to publish, or preparation of the manuscript." If this statement is not correct you must amend it as needed. Please include this amended Role of Funder statement in your cover letter; we will change the online submission form on your behalf.

Response 2: Thanks for these suggestions. We have attached a revised role of funder statement to the cover letter.

Q3. We note that you have provided funding information that is not currently declared in your Funding Statement. However, funding information should not appear in the Acknowledgments section or other areas of your manuscript. We will only publish funding information present in the Funding Statement section of the online submission form. Please remove any funding-related text from the manuscript and let us know how you would like to update your Funding Statement. Currently, your Funding Statement reads as follows: "This paper is funded by Anhui provincial key clinical specialties of the 14th Five-Year Plan (2021-25)."

Response 3: Thanks for these suggestions. We have made revisions to the acknowledgments section to meet the requirements.

Q4. We note that your Data Availability Statement is currently as follows: [All relevant data are within the manuscript and its Supporting Information files.]

Please confirm at this time whether or not your submission contains all raw data required to replicate the results of your study. Authors must share the “minimal data set” for their submission. PLOS defines the minimal data set to consist of the data required to replicate all study findings reported in the article, as well as related metadata and methods (https:// journals. plos.org /plosone /s/data-availability#loc-minimal-data-set-definition).

Response 4: Thanks for these suggestions. We have made the modifications as requested.

Q5. Please review your reference list to ensure that it is complete and correct. If you have cited papers that have been retracted, please include the rationale for doing so in the manuscript text, or remove these references and replace them with relevant current references. Any changes to the reference list should be mentioned in the rebuttal letter that accompanies your revised manuscript. If you need to cite a retracted article, indicate the article’s retracted status in the References list and also include a citation and full reference for the retraction notice.

Response 5: Thanks for these suggestions. We have reviewed all the references in the list, and they meet the required standards.

Dear Reviewer 1,

Thank you very much for your time involved in reviewing this manuscript. We have studied your comments carefully and carefully revised the manuscript as suggested.

Q1. Subtypes were ambiguously defined.

Enrichment analysis linked IM1-IMD subtypes with IL-8 and TNF pathways, IM2-IMB with leukocyte activation and NK cell cytotoxicity, IM3-IMC with mitotic processes, and IM4-IMA with CNS and muscle development. These associations are descriptive and lack unified biological significance, let alone clinical relevance.

Clustering analysis from figures demonstrates unclear differences among groups, and survival analyses lack reproducibility.

Response 1: Thanks for these suggestions. For the definition of each subtype, we first analyzed the clinical prognosis and immune infiltration characteristics of each subtype; then, based on the differential expression genes (DEGs) of each subtype, enrichment analysis was performed to obtain the biological processes that each subtype may be involved in; finally, the clinical molecular characteristics of each subtype were analyzed. This study defines different subtypes based on prognosis, immune infiltration characteristics, enrichment analysis, and clinical molecular characteristics. At the same time, this study included gliomas from both the CGGA and TCGA databases for mutual verification. Of course, more genomic data analysis and relevant molecular biology experiments will be needed in the future to validate these conclusions. Hope this explanation has addressed your concerns.

Dear Reviewer 2,

Thank you very much for your time involved in reviewing this manuscript. We are very grateful for your evaluation and recognition of this study.

Minor issues:

Q1. The manuscript mentions the use of the CGGA and TCGA datasets, but lacks detail on how these datasets were selected, processed, and used in analyses. Please include a detailed description of the datasets, such as the types of data available (e.g., RNA-seq, clinical data), and specific cohorts.

Response 1: Thanks for these suggestions. We have added description of the datasets in Materials and Methods as you suggested (Line 171-177). Hope the adjustment has addressed your concern.

Q2. Readers may need more information about the normalization methods for the transcriptomic data.

Response 2: Thanks for the comment. We have added description of the datasets in Materials and Methods as you suggested (Line 177-179). Hope this response has addressed your concern.

Q3. Please consider describing how the differentially expressed genes (DEGs) were selected, including the criteria for differential expression such as statistical thresholds (e.g., p-value, fold change) and the tools used.

Response 3: Thanks for this suggestion. We have added description of the selection criteria for DEGs in Materials and Methods as you suggested (Line 186-189). Hope this response has addressed your concern.

Q4. The manuscript briefly mentions using DAVID for GO and KEGG pathway analyses without details on the processes involved. Please consider mentioning more of the DAVID tool overview and specific settings applied.

Response 4: Thanks for this suggestion. We have added description of the DAVID tool in Materials and Methods as you suggested (Line 196-203). Hope this response has addressed your concern.

Q5. In the part of the discussion related to proposed strategies for different immune subtypes, it would be beneficial to directly refer to the relevant medical literature to stronger justify the treatment recommendations.

Response 5: Thanks for this constructive comment. This detailed description was added to Line 387-402. Hope this response has addressed your concern.

Q6. Please acknowledge the study limitations and suggest future research directions.

Response 6: Thanks for this valuable suggestion. This detailed description was added to Line 421-425. Hope the adjustment has addressed your concern.

Again, we highly appreciate your comments for this manuscript and wish these explanations and corrections have addressed your concerns.

---

## [Editor Report · Decision Letter 1]

30 Sep 2024

Machine Learning–Based New Classification for Immune Infiltration of Gliomas

PONE-D-24-22148R1

Dear Dr. Yuan,

We’re pleased to inform you that your manuscript has been judged scientifically suitable for publication and will be formally accepted for publication once it meets all outstanding technical requirements.

Kind regards,

Michael C Burger, M.D.

Academic Editor

PLOS ONE
---

## [Editor Report · Acceptance letter]

16 Oct 2024

PONE-D-24-22148R1 

PLOS ONE

Dear Dr. Yuan, 

I'm pleased to inform you that your manuscript has been deemed suitable for publication in PLOS ONE. Congratulations! Your manuscript is now being handed over to our production team.

Kind regards, 

on behalf of

Dr. Michael C Burger 

Academic Editor

PLOS ONE